# Insecticidal Activity of Plant Powders against the Parasitoid, *Pteromalus venustus*, and Its Host, the Alfalfa Leafcutting Bee

**DOI:** 10.3390/insects11060359

**Published:** 2020-06-09

**Authors:** Mikhaela Ong, Nora Chomistek, Hanna Dayment, Wayne Goerzen, Danica Baines

**Affiliations:** 1Lethbridge Research and Development Centre, Agriculture and Agri-Food Canada, Lethbridge, AB T1J 4B1, Canada; mikhaelaong@gmail.com (M.O.); nora.chomistek@canada.ca (N.C.); hannadayment@gmail.com (H.D.); 2LCB Research LLC, Saskatoon, SK S7K 5A4, Canada; goerzenw@lcbresearch.com

**Keywords:** plant powders, parasitoid, Alfalfa leafcutting bee, toxicity

## Abstract

Developing a bee-friendly alternative to traditional insecticides used within commercial environments can contribute to reductions in pesticide exposure experienced by managed bees. We performed acute contact toxicity studies using fifteen plant powders from seven plant families against a parasitoid pest, *Pteromalus venustus*, and its host, the Alfalfa leafcutting bee (ALB). Ajwain, cinnamon, clove, cumin, fennel, ginger, nutmeg, oregano and turmeric applied at low contact concentrations had sufficient fumigant properties to cause equivalent or higher parasitoid mortality as that obtained with the traditional insecticide. Nutmeg adversely affected adult ALBs at both low and high contact concentrations, thus eliminating it as a candidate. Increasing the contact concentrations did not consistently increase parasitoid control but did increase adverse effects on the ALBs. In addition, the efficacious plant powders significantly reduced the sexual function and fertility of the female parasitoids, a feature not associated with the traditional insecticide. The dual nature of the mechanisms underlying the effects of the plant powders may translate into effective control of the parasitoid populations in the commercial environment. The results reported here support further evaluations of Ajwain, cinnamon, clove, cumin, fennel, ginger, oregano and turmeric as potential botanical insecticides for control of *P. venustus*.

## 1. Introduction

Alfalfa leafcutting bees (ALBs, *Megachile rotundata* Fab.) are gregarious, solitary bees that are important pollinators of alfalfa and canola. Each year, about 4 billion ALB adults are released into alfalfa and canola fields in western Canada and the northwestern US to pollinate the flowers, maximizing seed production [1,2]. Shelters containing nest blocks are provided within the fields for the mated female bees to construct and provision their nests. To do this, they gather leaves, nectar and pollen for individual nest cells, laying one egg in each cell after depositing the provisions and finally enclosing and capping off the developing larvae within the leaf-based cocoon. At the end of the summer, the nest blocks containing the cocoons are collected from the fields, dried for a period, and then harvested before being stored until the following spring. At that time, the cocoons are added to trays and transferred to an incubator maintained under standard ALB rearing conditions. The trayed cocoons are incubated for an average of 21 days, where male bees emerge between day 18 and 28, while the female bees emerge between day 21 and 28 [3]. The newly emerged bees may be held in the cold to synchronize emergence with crop phenology or transported directly to the field and released. Trays remain within the field for 7 to 10 days to ensure that female emergence is complete and all bees have left the trays. 

*Pteromalus venustus* Walker (Hymenoptera: Pteromalidae) is a common parasitoid of the developing ALBs [4,5,6,7]. This parasitoid is present in the alfalfa and canola fields during the summer and is passively transported within the nest blocks from the fields in late summer [1,2]. During the fall cocoon drying process, female parasitoids are able to lay eggs and if the conditions are right, continue during the storage period. Each female parasitoid can oviposit an average of 26 eggs into one cocoon and can lay up to 110 eggs [1], causing the loss of an average of four developing ALBs. In the spring incubation of the trayed cocoons, the developing parasitoids begin emerging as adults on day 8 or 9, giving them ample time to parasitize the remaining developing ALB prepupae. The actions of the first generation of parasitoids, left uncontrolled, translates into a 30-fold increase in the number of second-generation parasitoids, amplifying bee losses from parasitism [5,8]. An organophosphate insecticide, Vapona^®^ (Dichlorvos), is applied between day 7 and 13 during the spring incubation to reduce the rate of parasitism of the first generation of parasitoids by about 67% [8]. However, the vapor phase of this chemical poses a significant risk to the health of the applicators [9] and emerging bees [8,10], providing the incentive to develop safer alternatives, such as botanical insecticides. 

Many studies have evaluated the insecticidal properties of plant powders from seeds, leaves, flower buds, flower heads, rhizomes, bark and peels to protect stored products from a variety of pest insects [11,12,13,14,15,16,17,18,19,20,21,22,23,24,25,26,27,28,29]. The essential oils (EOs) released from the plant powders act as fumigants, repellants and reproductive antagonists, resulting in significant reductions in the target pest populations. Each plant family has a unique combination of EOs that together function to mediate the effects against the pest insects, with plant powders having lower efficacy than their corresponding EO mixtures [13,14,15,23]. To our knowledge, no plant powders have been applied to control parasitoids that are the pests of a stored insect such as the ALB prepupa. Instead, as part of developing an integrated pest management (IPM) program for controlling stored product pests of grains, the impact of plant powders on one of the biocontrol agents, beneficial parasitoids or wasps, was evaluated. A diverse group of plant powders or their EOs from eight families (Apiaceae [30], Euphorbiaceae [25], Lamiaceae [22,31,32,33,34], Leguminosae [35], Myrtaceae [21], Poaceae [22,24], Solanaceae [35] and Zingiberaceae [23,34]), adversely affected beneficial parasitoids by increasing adult mortality, reducing the rate of parasitism, repelling the parasitoid, reducing adult longevity and reducing the emergence of the developing parasitoids. This suggests that plant powders may also be efficacious for *P. venustus*.

This study aimed to identify plant powders with insecticidal activity against adult *P. venustus* without eliciting adverse effects on the emerging adult ALBs. We first investigated the acute contact toxicity of commercially available plant powders (Ajwain, basil, cinnamon, clove, coriander, cumin, fenugreek, fennel, ginger, nutmeg, oregano, rosemary, sage, thyme and turmeric) belonging to seven plant families (Apiaceae, Lamiaceae, Lauraceae, Fabaceae, Myristicaceae, Myrtaceae and Zingiberaceae) against newly-emerged adult *P. venustus*. This was performed using the two types of contact encountered within treated trays: (1) low contact or a concentration equivalent to sparse adherence to the insect body from walking within the cocoon matrix, and (2) high contact or a concentration equivalent to heavy adherence to the insect body from interactions with accumulated plant powders during emergence or while walking within the cocoon matrix. Next, we examined whether pre-exposure of the parasitoid to the efficacious plant powders was associated with reproductive toxicity. Finally, to ensure that there were no negative impacts on adult bees, particularly the male bees, which emerge first and have a greater exposure period than female bees within the trays, we also examined the acute contact toxicity of the plant powders against adult male ALBs.

## 2. Materials and Methods

### 2.1. Rearing of ALBs

ALBs were reared as described previously [36]. Briefly, overwintered cocoon matrix from Alberta was provided by the Canadian Cocoon Testing Centre (Brooks, Alberta) and from commercial farmers. About 200 g of cocoon matrix (developing bee prepupae within cocoons, pollen balls, leaf material, dead bees within cocoons) were placed in plastic boxes (35 × 19 cm) with a metal screened lid for ventilation and Sefar^®^ e’mesh (Basic Mesh-123 Mesh 70 Micron PW; Davis International, Fairport, New York, NY, USA) to contain any emerging parasitoids. The boxes were maintained at 28 °C, 16:8 photoperiod and 50% relative humidity for 30 days, the standard incubation period. At day 15, two water-soaked pieces of paper towel placed within a weigh boat were added to the box to increase the relative humidity. Peak male adult emergence fell within 17 to 24 days of the 30-day incubation period. The adult male bees were transferred from the boxes using feather light forceps into 50 mL Fisherbrand™ conical tubes (Fisher Scientific Company, Ottawa, Ontario, Canada) just prior to use in the bioassays.

### 2.2. Rearing of Parasitoids

Stored cocoon matrix (~156 g) was transferred to 1000 mL transparent polypropylene containers. The central part of the lid had metal screening to allow for ventilation, and Sefar^®^ e’mesh was glued over the screening to contain any emerging parasitoids. A range of 30 to 50 female parasitoids were added to each container from an initial starter colony set up in a similar manner, the lid was replaced and then it was maintained at 28 °C, 16:8 photoperiod and 50% relative humidity for 15 days. Adult *P. venustus* were collected using an aspirator system (Tetra^®^ Whisper 60 Air Pump; PetSmart, Lethbridge, AB, Canada) with tubing connectors attached to a Fisherbrand™ specimen container (Fisher Scientific Company, Ottawa, ON, Canada) between day 12 and 15, just prior to use in the bioassays.

### 2.3. Plant Powders

Fifteen plant powders representing seven plant families were selected based upon the reported activity for pest insects and their local availability for farmers including ground Ajwain seed (Suraj^®^, No Frills, Lethbridge, AB, Canada), ground basil leaves (Western Family™, Save-On Foods, Lethbridge Alberta, Canada), cinnamon powder (No Name^®^, No Frills, Lethbridge, AB, Canada), clove powder (Bulk Barn^®^, Lethbridge, AB, Canada), coriander powder (McCormick^®^, Safeway, Lethbridge, AB, Canada), cumin powder (No Name^®^), ground Fenugreek seed (Suraj^®^), ground fennel seed (Suraj^®^), ginger powder (Western Family™), nutmeg powder (No Name^®^), ground oregano leaves (Western Family™), ground rosemary leaves (Western Family™), sage powder (No Name^®^), ground thyme leaves (No Name^®^) and turmeric powder (Quality^®^, Canadian Wholesale Club, Lethbridge, AB, Canada).

### 2.4. Acute Contact Toxicity—Adult P. venustus

A preliminary examination of the distribution of 1% plant powder within the cocoon matrix (*w*/*w*) in commercial trays and bioassay containers suggested that the adult parasitoids could come into contact with either an electrostatic amount of particles (low contact) from emergence and walking within the cocoon matrix, or a thick coating of particles (high contact) from settled plant powders within the cocoon matrix. All plant powders exhibited this uneven distribution within the cocoon matrix. To mimic those conditions, we physically coated adult parasitoids with low or high contact concentrations of each plant powder. This would provide us with insight into the dose range necessary to achieve control in the commercial tray environment. We also calculated the powder concentrations relative to the container volume, a method used to compare the released EO vapor potency among plant powders.

#### 2.4.1. Low Contact with Plant Powders

A mixture of ten male and female parasitoids were collected as described in Section 2.2. The lid was removed from a 90 mm Phoenix Biomedical Petri dish lined with Whatman^®^ (Fisher Scientific Company, Ottawa, Ontario, Canada) 85 mm filter paper that was previously dusted with a plant material using a stainless-steel sieve (8 cm). The parasitoids were added to the dish. The exposure concentration for each plant material in the bioassay container was 26 mg/cm^2^ of Ajwain seed, 22 mg/cm^2^ ground basil leaves, 28 mg/cm^2^ cinnamon powder, 19 mg/cm^2^ clove powder, 33 mg/cm^2^ coriander powder, 62 mg/cm^2^ cumin powder, 47 mg/cm^2^ ground Fenugreek seed, 21 mg/cm^2^ ground fennel seed, 79 mg/cm^2^ ginger powder, 47 mg/cm^2^ nutmeg powder, 22 mg/cm^2^ ground oregano leaves, 16 mg/cm^2^ ground rosemary, 20 mg/cm^2^ sage powder, 22 mg/cm^2^ ground thyme leaves and 29 mg/cm^2^ turmeric powder. The variation in the amounts applied reflects the differences in particle size and how it affects the distribution on the filter paper. Honey water (1:1) was supplied ab libitum via a cotton pad, with no visible contamination from the plant powders transferred to the feeding surface. The lid was then replaced and taped to the bottom of the dish to prevent the adult parasitoids from escaping. Untreated parasitoids were handled in the same manner, but in the absence of a plant powder. Normally, the untreated parasitoids will pick up an electrostatic concentration of leaf powder as they emerge from the cocoons. This served as the negative control. The bioassay containers were transported to an incubator and maintained at 28 °C, 16:8 photoperiod and 50% relative humidity. Mortality was assessed at 24, 48, 72 and 96 h. Ten adult *P. venustus* were weighed before and after exposure to the plant powders. No weight changes were observed per individual adult, but sparse plant particles were visible on the parasitoids when examined with a dissecting microscope. The experiment was replicated six times using cocoons obtained from different regions within Alberta.

#### 2.4.2. High Contact with Plant Powders

A mixture of ten male and female parasitoids were collected as described in Section 2.2 and transferred to a 1.5 mL Axygen^®^ (Fisher Scientific Company, Ottawa, ON, Canada) tube containing 0.1 g of a plant powder. The exposure concentration for each plant material in the bioassay container was equivalent to 589 mg/cm^2^, or about 20 times the concentration encountered for the low contact with plant powders in Section 2.4.1. The tube was closed and gently rolled by hand to distribute the plant material onto the parasitoids. The tube contents were poured into a 90 mm Phoenix Biomedical Petri dish lined with Whatman^®^ 85 mm filter paper. Honey water (1:1) was supplied ab libitum via a cotton pad, with no visible contamination from the plant powders transferred to the feeding surface. The lid was then replaced and taped to the bottom of the dish to prevent the parasitoids from escaping. Untreated parasitoids were handled in the same manner, but in the absence of plant powder. The bioassay containers were transported to an incubator and maintained at 28 °C, 16:8 photoperiod and 50% relative humidity. Mortality was assessed at 24, 48, 72 and 96 h. Ten adult *P. venustus* were weighed before and after exposure to the tube-dusting. No weight changes were observed, but adults were visibly coated with plant particles. The experiment was replicated six times using cocoons obtained from different regions within Alberta.

### 2.5. Reproductive Toxicity—P. venustus

Each efficacious plant powder, as determined in Section 2.4, was applied to the filter paper as described in Section 2.4.1, but at half to a quarter of the concentration to prevent female parasitoid mortality, or about 4.75 to 40 mg/cm^2^. Clove and turmeric were applied at the lowest concentration to prevent mortality within the study period. Ten female and two male parasitoids were collected as described in Section 2.2 and added to a Petri dish lined with the treated filter paper. Honey water (1:1) was supplied ab libitum via a gauze-covered Eppendorf tube that was taped to the filter paper. The lid was then placed on top and taped to the bottom to prevent the adult parasitoids from escaping. Untreated parasitoids were handled in the same manner, but in the absence of plant powder. The Petri dishes were transported to an incubator and maintained at 28 °C, 16:8 photoperiod and 50% relative humidity for 24 h. Then, each female parasitoid was transferred to the lid of a 112 mL disposable polypropylene dish fitted with four caps (VWR disposable/sterile 8-strip caps for library tubes; VWR International, Edmonton, AB, Canada) glued to the lid, each containing one ALB fifth-instar prepupa that had been removed from the cocoon. Honey water was supplied ab libitum by adding a 1 cm piece of cotton swab. The bottom of the dish was then snapped onto the lid. Untreated parasitoids were handled in the same manner, but in the absence of plant powder. The bioassay containers were transported to an incubator and maintained at 28 °C, 16:8 photoperiod and 50% relative humidity for 7 days. Each cup was then opened and assessed for the number of parasitized prepupa or the number of parasitoid larvae per prepupa. The experiment was replicated three times using parasitoids obtained from different regions within Alberta.

### 2.6. Acute Contact Toxicity—Adult ALBs

#### 2.6.1. Low Contact with Plant Powders

Ten adult male ALBs were collected, as described in Section 2.1, and then transferred to a 90 mm Phoenix Biomedical Petri dish lined with 85 mm Whatman^®^ filter paper that was previously dusted with a plant powder applied as described in Section 2.4.1. The Petri dishes were gently shaken, and the ALBs interacted with the plant powders for an additional 5 min. The adult bees were then cooled for 3 min at 4 °C and finally, transferred to a 112 mL polypropylene dish with two 2.5 × 2.5 cm coarse floor vent filters that served as perches. The lid of the plastic container was customized with a metal screened lid to provide ventilation and a port to hold an inverted 5 mL Eppendorf^®^ tube (Fisher Scientific Company, Ottawa, ON, Canada). This tube contained 1:1 honey water and was covered with layered cotton gauze held in place with an elastic. Untreated bees were handled in the same manner except no plant powder was applied to the filter paper. The bioassay containers were transported to an incubator and maintained at 28 °C, 16:8 photoperiod and 50% relative humidity. Mortality was assessed at 24, 48, 72 and 96 h. Individual adult bees were weighed before and after exposure. The mean weight of the plant material picked up by an adult bee ranged between 0.5 and 1 mg per bee. Since 10 bees were added to each container, this represents a range between 5 and 10 mg per container, or 32 to 65 µg/cm^2^. The experiment was replicated six times using cocoons obtained from different regions within Alberta.

#### 2.6.2. High Contact with Plant Powders

Ten adult male ALBs were collected as described in Section 2.1 and then added to a 50 mL Fisherbrand™ conical tube containing 2 g of plant material. The lid was placed onto the tube and gently rolled four to six times. The visibly coated bees were then transferred to a 112 mL plastic container with two 2.5 × 2.5 cm coarse floor vent filters that served as a perch. The lid of the plastic container was customized with a metal screened lid to provide ventilation and a port to hold an inverted 5 mL Eppendorf^®^ tube. The 5 mL tube contained 1:1 honey water and was covered with layered cotton gauze held in place with an elastic. Untreated bees were handled in the same manner except no plant powder was applied to the filter paper. The bioassay containers were transported to an incubator and maintained at 28 °C, 16:8 photoperiod and 50% relative humidity. Mortality was assessed at 24, 48, 72 and 96 h. Individual adult bees were weighed before and after exposure to a tube-dusting of each plant material. The mean weight of the plant material picked up by an adult bee ranged between 3 and 5 mg per bee. Since 10 bees were added to each container, this represents a plant powder concentration between 30 and 50 mg per container, which is 0.20 to 0.33 mg/cm^2^ or about 6 times higher than the amount applied in the low contact bioassays, Section 2.6.1. The experiment was replicated six times using cocoons obtained from different regions within Alberta.

### 2.7. Statistical Analysis

The count data were analyzed using the Loglinear component within SYSTAT 13.0 (Systat Software, Inc., San Jose, CA, USA), with two discrete variables, treatment and time-period. We examined the test for Model terms panel comparing the likelihood-ratio chi-square for the full model to the same value for the smaller model. To determine whether the removal of a term results in a significant decrease in the fit, we then looked at the difference in these statistics. If the fit was worsened with the removal of the term, it remained in the model. With each change in the model, we checked Raftery’s BIC (Bayesian Information Criterion), which when negative, concludes that the model is preferable to the saturated model. For each factor in the model, z-scores were provided within the analysis, and the probabilities associated with the z-scores were obtained using z-tables. The results are expressed as a treatment mean or the effect of the treatments on a measured count variable, and this is compared with the effects of individual treatments.

## 3. Results

All results are provided as: name of plant powder, mean number of dead parasitoids/bees, z-score and *p*-value for the z-score.

### 3.1. Acute Contact Toxicity—Adult P. venustus

#### 3.1.1. Low Contact with Plant Powders (19 to 79 mg/cm^2^)

There was a significant effect of time (χ^2^ = 39.943, df = 3, *p* < 0.005) and treatment (χ^2^ = 927.039, df = 15, *p* < 0.005) on adult parasitoid mortality. There was a significantly lower parasitoid mortality recorded at 24 h of exposure (7.8, −5.332, *p* < 0.0002), followed by a significant increase in parasitoid mortality between 72 (8.3, 2.338, *p* = 0.0096) and 96 h (8.4, 4.334, *p* < 0.0002, Figure 1, Figure 2 and Figure 3). For untreated parasitoids (1.5, −5.077, *p* < 0.0002), the mean number of dead parasitoids was significantly lower (82%) than the treatment mean of 8.2. For the family Lamiaceae, the mean number of dead parasitoids for basil (3.0, −5.472, *p* < 0.0002), rosemary (2.6, −7.290, *p* < 0.0002), sage (4.0, −4.675, *p* < 0.0002) and thyme (3.4, −6.046, *p* < 0.0002) were significantly lower than the treatment mean, indicating that they have no significant effect on the adult parasitoids (Figure 1). As a result, these plant powders can be excluded as having adequate fumigant properties for activity against *P. venustus*. However, one member of the Lamiaceae, oregano (6.1, 2.023, *p* = 0.0217), significantly increased the mean number of dead parasitoids, achieving equivalent control as described for Vapona^®^ applications (67%) [8]. For the Apiaceae, several plant powders significantly increased parasitoid mortality compared with the treatment mean, including Ajwain (9.7, 12.455, *p* < 0.0002), cumin (7.1, 4.598, *p* < 0.0002) and fennel (10, 13.390, *p* < 0.0002; Figure 2). These plant powders are fast acting, achieving greater than 60% mortality at 24 h. In contrast, the mean number of dead parasitoids was significantly lower than the treatment mean for coriander (3.1, −5.177, *p* < 0.0002), indicating that it has no effect on the adult parasitoids. Members of several other plant families, Lauraceae (cinnamon, 9.1, 10.805, *p* < 0.0002), Myristicaceae (nutmeg, 9.6, 11.939, *p* < 0.0002), Myrtaceae (clove, 10, 13.474, *p* < 0.0002) and Zingiberaceae (ginger, 7.9, 5.851, *p* < 0.0002; turmeric, 9.1, 10.185, *p* < 0.0002) significantly increased the mean number of dead parasitoids compared with the treatment mean (Figure 3). Cinnamon, clove, nutmeg and turmeric are also fast acting, achieving >60% mortality at 24 h. One member of the Fabaceae, Fenugreek (3.1, −5.858, *p* < 0.0002), caused significantly lower parasitoid mortality than the treatment mean, indicating that it has no effect on the adult parasitoids. The hierarchy for relative potency of the plant powders is Ajwain, cinnamon, clove, fennel, nutmeg, turmeric > cumin, ginger > oregano. The fast-acting nature of Ajwain, cinnamon, clove, cumin, fennel, nutmeg and turmeric suggests that these are the best candidates for replacing Vapona^®^ applications in commercial ALB operations.

To mimic the concentration used in the low contact studies, the amount of plant powders required per tray, using the cocoon weight occupying the tray volume, was calculated to be equal to 1 to 6% concentration, depending upon the plant powder.

#### 3.1.2. High Contact with Plant Powders (589 mg/cm^2^)

There was a significant effect of time (χ^2^ = 71.885, df = 3, *p* < 0.005) and treatment (χ^2^ = 711.646, df = 15, *p* < 0.005) on adult parasitoid mortality. There was a significantly lower parasitoid mortality recorded at 24 h of exposure (6.2, −7.243, *p* < 0.0002) followed by a gradual significant increase in parasitoid mortality at 72 (7.4, 3.056, *p* = 0.0011) and 96 h (7.6, 5.628, *p* < 0.0002; Figure 4, Figure 5 and Figure 6). Increasing the contact concentration of the plant powders did not alter the dose-response relationship observed for parasitoid mortality. For untreated parasitoids (1.6, −4.539, *p* < 0.0002), the mean number of dead parasitoids was significantly lower (78%) than the treatment mean of 7.2. For the family Lamiaceae, the mean number of dead parasitoids for basil (4.5, −4.316, *p* < 0.0002), rosemary (2.6, −5.808, *p* < 0.0002), sage (2.1, −6.002, *p* < 0.0002) and thyme (2.9, −4.316, *p* < 0.0002) was significantly lower than the treatment mean (Figure 4). As a result, these plant powders can be excluded as having adequate fumigant properties for activity against *P. venustus*. In contrast to the low contact application, oregano (6.4, 1.421, *p* = 0.0823) did not significantly increase the number of dead parasitoids compared with the treatment mean, but did result in significantly higher parasitoid mortality (64%) compared with the untreated parasitoids (16%). For the family Apiaceae, several plant powders significantly increased parasitoid mortality compared with the treatment mean, including Ajwain (7.4, 4.295, *p* < 0.0002), cumin (6.4, 3.861, *p* < 0.0002) and fennel (9.2, 13.650, *p* < 0.0002; Figure 5). These plant powders are fast acting, achieving >60% mortality at 24 h. In contrast, the mean number of dead parasitoids was significantly lower (65%) than the treatment mean for coriander (2.5, −5.906, *p* < 0.0002), indicating that it has no effect on the adult parasitoids. Members of several other plant families, Lauraceae (cinnamon, 7.5, 7.659, *p* < 0.0002), Myristicaceae (nutmeg, 8.3, 11.027, *p* < 0.0002), Myrtaceae (clove, 9.8, 15.446, *p* < 0.0002) and Zingiberaceae (ginger, 6.8, 5.047, *p* < 0.0002; turmeric, 7.6, 7.761, *p* < 0.0002) significantly increased parasitoid mortality compared with the treatment mean (Figure 6). Only clove was fast acting, achieving >60% mortality at 24 h. One member of Fabaceae, Fenugreek (3.2, −2.484, *p* = 0.006), significantly decreased parasitoid mortality compared with the treatment mean, indicating that it has no effect on the adult parasitoids. The hierarchy for relative potency of the plant powders is clove, fennel, nutmeg > Ajwain, cinnamon, turmeric > cumin, ginger, oregano. The change in the hierarchy from the low contact exposure suggests that clove, fennel and nutmeg have a higher essential oil content that facilitates the recorded higher parasitoid mortality. Only four plant powders act quickly (24 h) under high contact conditions to elicit parasitoid mortality, including Ajwain, clove, cumin and fennel.

Exposure of the adult parasitoids to nine of the fifteen plant powders elicited a mean mortality of 60% or greater for adult *P. venustus*.

### 3.2. Reproductive Toxicity (4.75 to 40 mg/cm^2^)—P. venustus

There was a significant effect of treatment (χ^2^ = 58.813, df = 8, *p* < 0.005) on the number of parasitized ALB prepupae (Figure 7). Untreated parasitoids (2, 7.847, *p* < 0.0002) parasitized significantly more ALB prepupae than the treatment mean of 0.36. In contrast, Ajwain (0.02, −1.319, *p* = 0.09342), cinnamon (0.05, −1.319, *p* = 0.09342), clove (0, −1.487, *p* = 0.06811), cumin (0.15, −0.985, *p* = 0.16109), ginger (0.23, −0.156, *p* = 0.44644), oregano (0, −1.487, *p* = 0.06811) and turmeric (0.3, 0.336, *p* = 0.3669) applications were equivalent to the treatment mean, parasitizing 82% lower ALB prepupae than with the untreated parasitoids. Although fennel (0.55, 1.623, *p* = 0.0526) significantly increased the number of parasitized ALB prepupae compared with the treatment mean, it was 72% lower than with the untreated parasitoids. This suggests that the plant powders change the sexual function of the female parasitoids, resulting in fewer attacks.

There was a significant effect of treatment (χ^2^ = 558.657, df = 8, *p* < 0.005) on the number of parasitoid larvae developing from each parasitized ALB prepupae (Figure 7). Untreated parasitoids (18.64, 24.632, *p* < 0.0002) had significantly more parasitoid larvae develop per ALB prepupae compared with the treatment mean of 3.29. Ginger applications (1.62, −0.508, *p* = 0.30503) elicited an equivalent number of parasitoid larvae as the treatment mean, or were 91% lower than with the untreated parasitoids. In contrast, Ajwain (0.23, −3.322, *p* = 0.00045), cinnamon (1.20, −2.461, *p* = 0.00695), clove (0, −4.178, *p* < 0.00003), cumin (1.90, −1.970, *p* = 0.02442) and oregano (0, −4.178, *p* < 0.00003) applications elicited a significantly lower number of parasitoid larvae than the treatment mean, or were 94% lower than with the untreated parasitoids. Although fennel (3.57, 4.677, *p* < 0.0002) and turmeric (2.7, 2.667, *p* = 0.0038) significantly increased the number of parasitoid larvae compared with the treatment mean, this increase was still significantly lower (81–86%) than with the untreated parasitoids. We removed the zero data points from this data set to determine if the differences in larval numbers were associated with the egg/larval parasitoid development. Untreated parasitoids laid eggs that developed into a mean of 22.9 parasitoids per ALB prepupae. In contrast, Ajwain, cumin, fennel and ginger weakly reduced (~45%) the fertility of the female parasitoids (14.0, 12.7, 17.3 and 13.9 parasitoids per bee larvae, respectively), while cinnamon, clove, oregano and turmeric strongly reduced (~85%) the fertility of the female parasitoids (3.4, 0.0, 0.0 and 6.2 parasitoids per bee larvae, respectively).

### 3.3. Acute Contact Toxicity—Adult Male ALB

#### 3.3.1. Low Contact with Plant Powders (32 to 65 µg/cm^2^)

There was a significant effect of time (χ^2^ = 220.019, df = 3, *p* < 0.005) and treatment (χ^2^ = 48.48, df = 15, *p* < 0.005) on adult male ALB mortality. There was a significantly lower adult ALB mortality recorded at 24 h of exposure (1, −6.094, *p* < 0.0002), followed by a significant increase in adult ALB mortality between 72 (1.9, 2.491, *p* = 0.0064) and 96 h (3.3, 13.179, *p* < 0.0002; Figure 8, Figure 9 and Figure 10). For untreated ALBs (1.8, −0.634, *p* = 0.2643), the mean number of dead adult ALBs was equivalent to the treatment mean of 2.8. For the family Lamiaceae, the mean number of dead adult ALBs for basil (4.1, −1.128, *p* = 0.1292), oregano (2.1, −0.634, *p* = 0.2643) and thyme (1.8, 0.834, *p* = 0.2033) was also equivalent to the treatment mean, indicating that they pose no risk for ALB losses (Figure 8). Interestingly, basil elicited highly variable mortality (0–9) resulting in a high mean mortality at 96 h, suggesting that there may be a genetic or management component to the regional differences in ALB sensitivity. In contrast, rosemary (4.2, 3.606, *p* < 0.0002) and sage (1.9, 2.019, *p* = 0.0217) significantly increased ALB mortality compared with the treatment mean. Since Vapona^®^ applications elicit about a 16% adult ALB loss [8], only rosemary poses a greater risk for ALB losses than the current method for controlling the parasitoid. For the family Apiaceae, several plant powders either significantly decreased adult ALB mortality or were equivalent to the treatment mean, including coriander (2.0, −1.863, *p* = 0.0314), cumin (1.8, −0.634, *p* = 0.2643) and fennel (1.4, −1.128, *p* = 0.1292; Figure 9). These powders, therefore, pose no risk for adult ALB losses. In contrast, Ajwain (2.5, 2.019, *p* = 0.0217) significantly increased the number of dead ALBs compared with the treatment mean, but was within the 16% losses reported for Vapona^®^ [8], warranting further investigation prior to eliminating this plant powder as a candidate for controlling the parasitoids. Members of several other plant families, Lauraceae (cinnamon, 2.2, 1.074, *p* = 0.1423), Fabaceae, (Fenugreek, 1.0, −2.970, *p* = 0.0015), Myrtaceae (clove, 5.6, 0.834, *p* = 0.2033) and Zingiberaceae (ginger, 2.4, 0.349, *p* = 0.3632; turmeric, 2.3, 1.074, *p* = 0.1423) significantly decreased ALB mortality or were not higher than the treatment mean (Figure 10). These plant powders also did not pose a risk for adult ALB losses. Clove elicited high ALB losses, but the effect was highly variable (0–9 dead) and was dependent upon the origin of the cocoons. In contrast, one member of the Myristicaceae family (nutmeg, 3.1, 3.385, *p* = 0.0003) significantly increased adult ALB losses compared with the treatment mean, eliciting about a 72% greater loss than that recorded for untreated ALBs. The hierarchy of the relative potency for the plant powders to harm ALBs is nutmeg, rosemary > clove. Within the ALB commercial industry, 16% ALB losses [8] are acceptable. Based upon our results, nutmeg and rosemary cause higher ALB losses than the acceptable commercial level, eliminating them as candidates for parasitoid control.

#### 3.3.2. High Contact with Plant Powders (0.20 to 0.33 mg/cm^2^)

There was a significant effect of time (χ^2^ = 97.508, df = 3, *p* < 0.005) and treatment (χ^2^ = 72.636, df = 15, *p* < 0.005) on adult male ALB mortality. There was a significant lower mean ALB mortality recorded at 24 h of exposure (1.6, −6.623, *p* < 0.0002), followed by a gradual significant increase in adult ALB mortality at 72 (2.9, 3.318, *p* = 0.0005) and 96 h (3.6, 8.243, *p* < 0.0002; Figure 11, Figure 12 and Figure 13). For untreated ALBs (1.0, −2.332, *p* = 0.0099), the mean number of dead adult ALBs was significantly lower (64%) than the treatment mean of 1.2. For the family Lamiaceae, the mean number of dead adult ALBs for basil (1.3, −1.379, *p* = 0.0838), oregano (1.7, −0.610, *p* = 0.2709), thyme (0.2, −1.762, *p* = 0.0392) and sage (1.8, −0.418, *p* = 0.3372) either significantly decreased ALB mortality or was equivalent to the treatment mean of 1.2 (Figure 11). In contrast, rosemary (2.5, 2.028, *p* = 0.0212) significantly increased adult ALB mortality compared with the treatment mean, and these losses were above those reported with Vapona^®^ applications [8]. For the family Apiaceae, Ajwain (1.3, −1.379, *p* = 0.0838) and coriander (1.7, −0.227, *p* = 0.4090) significantly reduced adult ALB mortality compared with the treatment mean (Figure 12). In contrast, cumin (2.8, 3.474, *p* = 0.0003) and fennel (3.2, 2.212, *p* = 0.0136) significantly increased the number of dead adult ALBs compared with the treatment mean, and these losses are above those reported with Vapona^®^ applications. These plant powders should not be applied at concentrations where adult ALBs can encounter accumulated material within the cocoon matrix. Members of several other plant families, Lauraceae (cinnamon, 1.2, −1.379, *p* = 0.0838), Fabaceae, (Fenugreek, 1.7, 0.155, *p* = 0.5636) and Zingiberaceae (ginger, 2.0, −0.227, *p* = 0.4090; turmeric, 1.3, −1.379, 0.0838), caused adult ALB mortality equivalent to the treatment mean (Figure 13), but were within the 16% bee loss attributed to Vapona^®^ applications [8]. In contrast, one member of the family Myristicaceae (nutmeg, 4.5, 6.711, *p* < 0.0002) and one member of the family Myrtaceae (clove, 2.8, 2.028, *p* = 0.0212) significantly increased adult ALB mortality compared with the treatment mean, and these losses are above those reported with Vapona^®^ applications [8]. As with the lower contact results, clove elicited high ALB losses at 96 h, but was highly variable, suggesting that there may be a genetic or management component to the regional differences in bee sensitivity. The hierarchy for the relative potency of the negative effects of the plant powders on adult ALBs is nutmeg > cumin, fennel, rosemary > clove. Only nutmeg poses a higher risk for ALB losses than the currently applied insecticide and, therefore, has been eliminated as a candidate for parasitoid control.

One of the factors affecting the outcome for contact toxicity with high concentrations or coatings of plant powders was related to bee behavior. The heavily coated bees tended to readily “buzz” the powder off the body within minutes of release into the bioassay container, and followed this with intensive grooming. The different particle sizes contributed to the times required for the “buzz” behavior to remove the particles from the bee’s body. Fine powders such as turmeric and cinnamon stayed on the body for the longest period and required intensive grooming, while course powders such as clove or rosemary were removed immediately followed by intensive grooming. Despite this, increasing the particle distribution on the bee body for oregano, sage, cumin, fennel, ginger, cinnamon and nutmeg increased either the mortality or the time required to elicit ALB mortality. This suggests that it is the EO content rather than the amount of powder deposited on the insect that is eliciting the effects on the adult ALBs. Since adult ALBs will come in contact with the plant powders as they emerge from and interact with the cocoon matrix during the spring incubation, we would suggest applications equivalent to the low concentration to ensure that the adult ALBs are not adversely affected.

Overall, the results suggest that clove, nutmeg and rosemary pose the highest risk to adult male ALBs, while Ajwain, cinnamon, fennel, turmeric, cumin, ginger and oregano are good candidates to apply to cocoons for reducing *P. venustus* populations.

## 4. Discussion

Plant powders and their corresponding EOs have been investigated extensively, with their biggest successes in controlling stored product pests [11,12,13,14,15,16,17,18,19,20,21,22,23,24,25,26,27,28,29]. In the current study, the stored commodity is not a grain but ALBs, and the pest insect is a parasitoid (*P. venustus*). This lends an added question as to the safety or risk for the commodity, developing ALBs. Current control strategies for *P. venustus* involve applying Vapona^®^ (active ingredient, Dichlorvos) at day 7 of the 30-day spring incubation period to overlap with the emergence of the first generation of *P. venustus* [3,7]. The Vapona^®^ is removed on day 13 and the incubators are vented to release any remaining vapor. This approach suppresses the rate of parasitism by the first generation of parasitoids by about 67% [8]. Concerns over the safety of the Vapona^®^ vapor for farmers in the confined incubators, together with the risk of the residues for the developing ALBs, provided the impetus to investigate alternative control measures for this parasitoid. Previous studies suggest that several EOs are harmful to beneficial parasitoids (Braconidae [21], Platygastridae [33], Pteromalidae [22,23,24,25,31,32,34], Trichogrammatidae [23,30,34,35]) applied in IPM programs for stored product pests. We investigated the contact toxicity of fifteen plant powders from seven plant families against adult *P. venustus*, but also examined whether exposure to the plant materials was safe for adult male ALBs. We identified nine plant powders that were toxic to adult *P. venustus*, including Ajwain, cinnamon, clove, cumin, fennel, ginger, nutmeg, oregano and turmeric. Of these, nutmeg and rosemary pose a high risk for adult male ALB toxicity, while clove poses a high risk for adult male ALB toxicity if the ALBs are from specific regions within Alberta. Therefore, nutmeg and rosemary should be eliminated from further studies, while clove should be investigated with caution.

Typically, the next phase of investigation for the efficacious plant powders is to extract the EOs and determine the toxicity against *P. venustus* and ALBs [12,19,20,29]. However, plant powders have several advantages over EOs within commercial ALB production; they are easy to obtain, cost effective, easy to incorporate at harvesting from the nest blocks, safe for the farmer, less toxic than EOs [11,12,13,14] and are safer for the developing ALBs. In addition, the EO vapors are more unstable than plant powders, undergoing oxidative damage, chemical transformations or polymerization reactions when exposed to heat, water, light or air [37,38]. Under the conditions employed during the spring incubation of the ALB cocoons, EOs would degrade rapidly (30 °C, 50% relative humidity), not penetrate the cocoon layer and potentially concentrate within the cocoons at the surface of the trays, thus harming the developing ALBs. The greater stability of the plant powders is a benefit and must be balanced by its lower potency when compared with EOs. A factor that must be addressed for commercial applications is the EO content of plant powders [12], which vary depending upon the plant source [13,23] and the manufacturing process [12,20]. This can be overcome at the commercial level by adding an additional step, testing the plant powder, prior to the large-scale spring incubation to ensure efficacy against the pest without harming the ALBs. Although this would be a challenge for some agricultural sectors, this is feasible for managers of ALBs.

To be effective fumigants, the EOs from the plant powders must vaporize under ALB spring incubation conditions. In this scenario, the released vapor can travel throughout the cocoon matrix, poisoning the parasitoid larvae or emerging adult parasitoids. In one earlier study, plant powders from the family Lamiaceae reduced the rate of parasitism or the reproductive activity of a beneficial Pteromalid wasp (24%), and reduced larval parasitoid development (28%), which is suggestive of reproductive toxicity and fumigant properties [31,32]. Many studies have monitored the fumigant properties of EOs against beneficial parasitoids. However, these studies used EOs from plant genera and/or species that are not available in Canada, except for basil [22], ginger [23,35], oregano [33,35] and thyme [33,35]. We demonstrated significant adult *P. venustus* mortality after exposure to oregano and ginger powders, but not with thyme or basil powders. EOs from several other plant families (Apiaceae, Lauraceae, Mytraceae, Zingiberaceae, Lamiaceae) are effective fumigants for beneficial parasitoids or Coleopteran pests of stored products, such as grains [11,12,13,14,15,16,17,18,19,20,21,22,23,24,25,26,27,28,29,30,31,32,33,34,35].

Exposure of Trichogramma wasps to the LC01 of an EO from a member of the Apiaceae family reduced female longevity to 4.5 days post-exposure compared with 10 days for untreated parasitoids [30]. In the current study, members of this family (Ajwain, cumin and fennel) elicited high toxicity for adult *P. venustus*, peaking between 3 and 4 days post-exposure and indicating that the plant powders are releasing sufficient EOs to act as effective fumigants. These powders had equivalent or higher toxicity against *P. venustus* compared with Vapona^®^ (67%), supporting their potential for commercial applications.

The EOs from several members of the Lamiaceae or Myrtaceae families cause high adult mortality for Braconid [21], Pteromalid [22,31,32], Platygastrid [33] and Trichogramma wasps [35]. We examined the impact of two other members of the Lamiaceae family, rosemary and sage plant powders, which have strong fumigant properties for pest insects [26]. These powders did not achieve equivalent mortality to Vapona^®^ applications, regardless of contact concentration, eliminating them as candidates for commercial applications. In contrast, a member of the Myrtaceae family, clove, was highly toxic to adult *P. venustus*, but it also harmed the adult ALBs, making this a less favorable candidate than other efficacious plant powders with no effect on the ALBs.

There is one study regarding the Zingiberaceae family and EO activity against beneficial wasps. Increasing concentrations of ginger oil (Zingiberaceae) cause significant larval and adult mortality of Anisopteromalus wasps, but had no effect on larval stages of a Trichogramma wasp [23]. We confirmed that ginger powder caused a level of toxicity for adult *P. venustus* equivalent to Vapona^®^, supporting its potential use in commercial applications. Interestingly, ginger oil extracted from a different plant species than that used in the current or an earlier study caused significant mortality of immature Trichogramma wasps [35], suggesting that the source of the plant powder or EO can affect outcomes.

There is no information regarding the fumigant properties of turmeric (Zingerberaceae) and cinnamon (Lauraceae) powder against beneficial parasitoids. However, the toxicity of turmeric and cinnamon powders was evaluated for several stored product pests [13,16,18,27] with highly variable results depending upon the stored product pest examined. A 1% cinnamon and turmeric powder application was ineffective against the rice weevil, but did cause significant mortality at 5% concentration, 90% and 32%, respectively [18]. In contrast, 2.5% cinnamon or turmeric powder applications elicit significant mortality for the cowpea weevil [18]. The unpredictable nature of the response is further illustrated by a very low concentration of 0.5% cinnamon powder, causing 100% Khapra and red flour beetle mortality [13]. Regardless, both cinnamon and turmeric powders elicit significant adult *P. venustus* mortality, about 20% higher than that achieved with Vapona^®^ applications. The variation in the reported responses to plant powders is suspected to be a function of whether a matrix, such as grain, is present or absent in the bioassay containers, with higher mortalities achieved in grain-free containers [12]. If we assume that the toxicity will decline within the mixed environment of the cocoon matrix, then only those plant powders with higher toxicity than Vapona^®^ applications should be considered for further investigation. Ajwain, cinnamon, clove, fennel, ginger and turmeric meet this criterion, with caution as previously noted regarding clove because of its adverse effects on ALBs.

Plant powders with fumigant and repellant properties are more effective against stored product pests than plant powders displaying only one mode of action [13,17,18,27]. Ajwain is a strong repellant for Trichogramma wasps [30]. For pest insects, cinnamon bark is a repellant for coleopteran beetles, while ginger rhizomes and stems are repellent for aphids [29]. Repellency is also measured by the effects of plant powders on the size of the F1 populations related to the rejection of the oviposition site and reproductive toxicity. A 5% concentration of cinnamon, clove, cumin, fennel, ginger and turmeric significantly reduces the F1 generation of rice weevil between 35 and 100% [16]. Similarly, 0.85% turmeric and cinnamon powder effectively reduces the F1 generation for the lesser grain borer [27]. Other pests, such as red flour beetle and Khapra beetle, require higher concentrations of cinnamon powder (1–5%) to significantly reduce the F1 generation [13]. In the current study, Ajwain, cinnamon, clove, cumin, fennel, ginger, oregano and turmeric are repellant as reflected in their reproductive toxicity for *P. venustus*.

Overall, the dual mechanisms displayed by Ajwain, cinnamon, cumin, fennel, ginger, oregano and turmeric against *P. venustus*, together with their safety for ALBs, lend further support for their potential use in protecting stored developing ALBs from parasitoid attacks during the spring incubation.

## 5. Conclusions

For over 20 years, an organophosphate insecticide, Vapona^®^, has been used to control adult *P. venustus* attacking developing ALBs during spring incubation in commercial operations. This study identified several candidate plant powders with dual mechanisms for targeting and reducing *P. venustus* populations that are safe for the adult male ALBs. Further studies are required to ensure their safety for female bees and to understand the mechanisms underlying the toxicity recorded for the efficacious plant powders.

## Figures and Tables

**Figure 1 insects-11-00359-f001:**
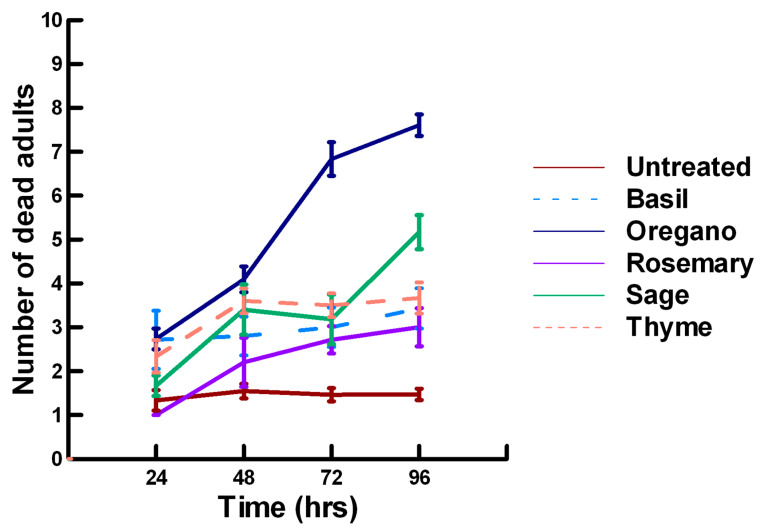
Acute contact toxicity with a low concentration of plant powders from the family Lamiaceae (basil, oregano, rosemary, sage and thyme) adherent to adult *Pteromalus venustus*. Means and standard error shown.

**Figure 2 insects-11-00359-f002:**
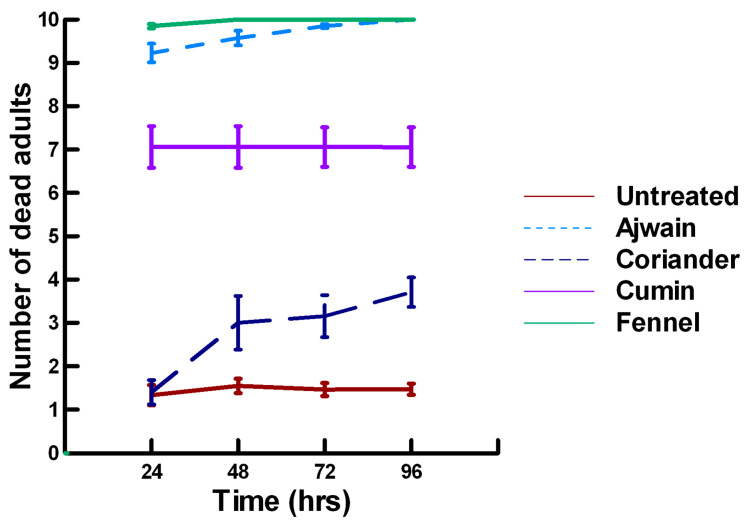
Acute contact toxicity with a low concentration of plant powders from the family Apiaceae (Ajwain, coriander, cumin and fennel) adherent to adult *Pteromalus venustus*. Means and standard error shown.

**Figure 3 insects-11-00359-f003:**
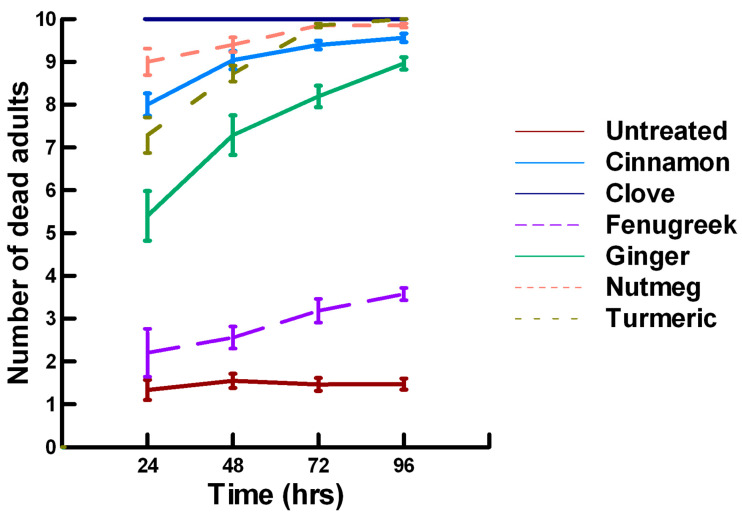
Acute contact toxicity with a low concentration of plant powders from the families Lauraceae (cinnamon), Myrtaceae (clove), Fabaceae (Fenugreek), Myristicaceae (nutmeg) and Zingiberaceae (ginger, turmeric) adherent to adult *Pteromalus venustus*. Means and standard error shown.

**Figure 4 insects-11-00359-f004:**
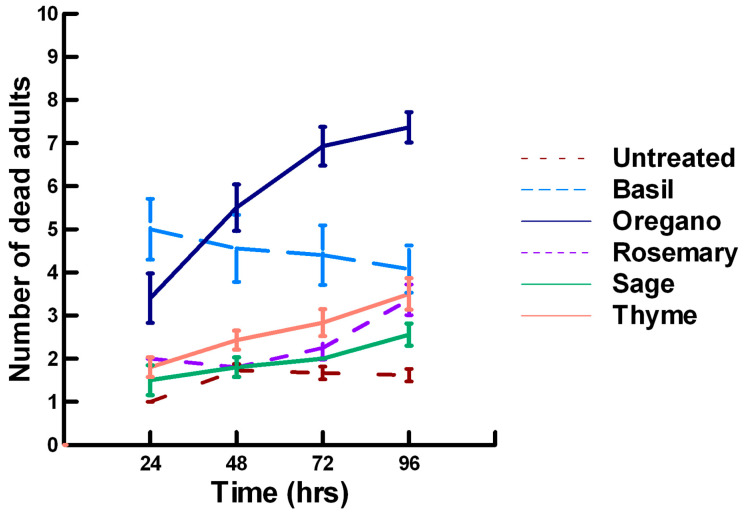
Acute contact toxicity with a high concentration of plant powders from the family Lamiaceae (basil, oregano, rosemary, sage and thyme) adherent to adult *Pteromalus venustus*. Means and standard error shown.

**Figure 5 insects-11-00359-f005:**
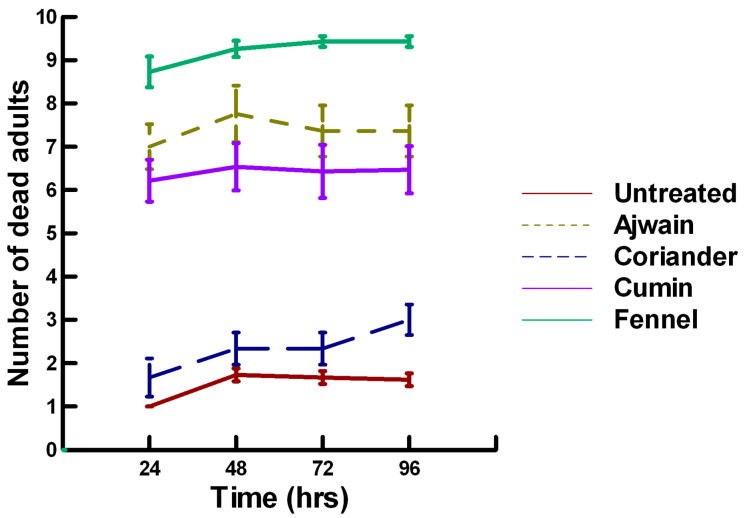
Acute contact toxicity with a high concentration of plant powders from the family Apiaceae (Ajwain, coriander, cumin and fennel) adherent to adult *Pteromalus venustus*. Means and standard error shown.

**Figure 6 insects-11-00359-f006:**
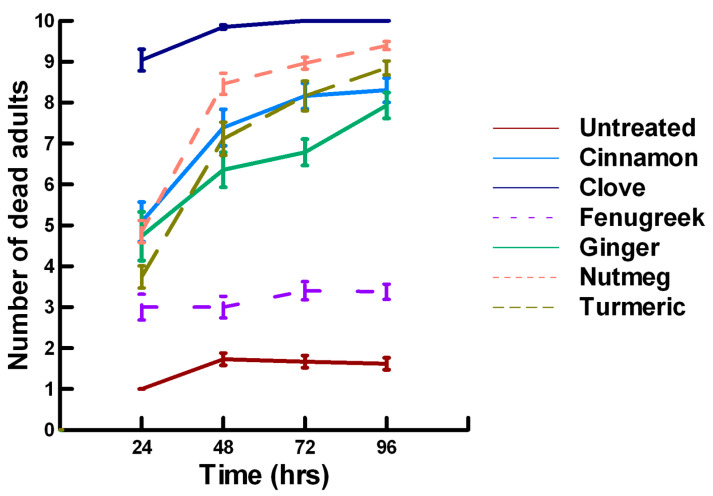
Acute contact toxicity with a high concentration of plant powders from the families Lauraceae (cinnamon), Myrtaceae (clove), Fabaceae (Fenugreek), Myristicaceae (nutmeg) and Zingiberaceae (ginger, turmeric) adherent to adult *Pteromalus venustus*. Means and standard error shown.

**Figure 7 insects-11-00359-f007:**
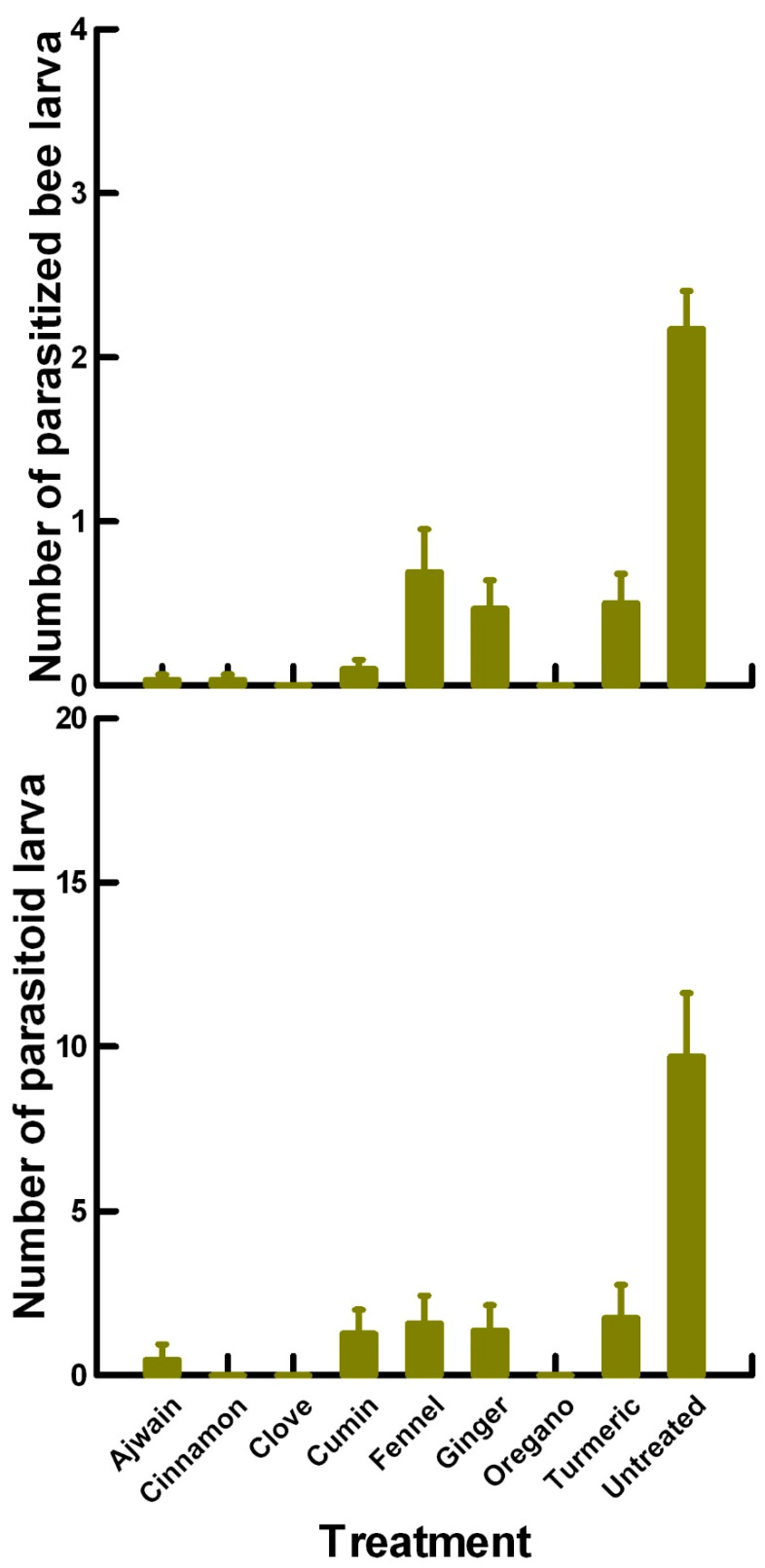
Reproductive toxicity of plant powders (Ajwain, cinnamon, clove, cumin, fennel, ginger, oregano and turmeric) to female *P. venustus* pre-exposed for 24 h. Means and standard error shown.

**Figure 8 insects-11-00359-f008:**
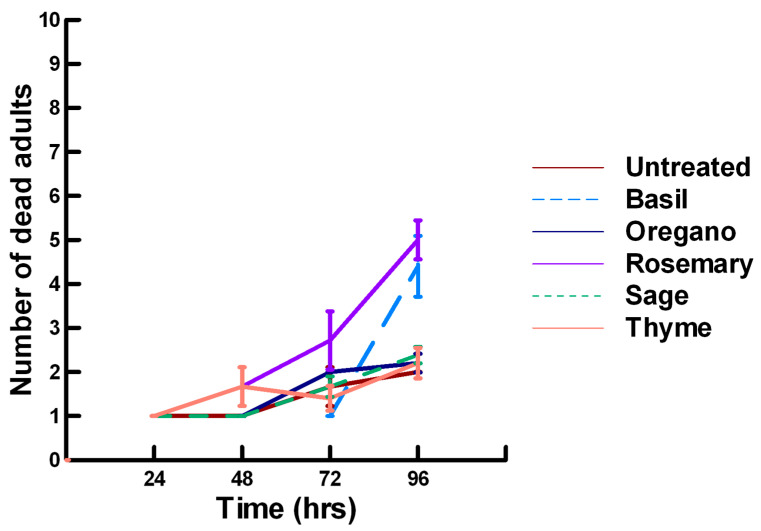
Acute contact toxicity with a low concentration of plant powders from the family Lamiaceae (basil, oregano, rosemary, sage and thyme) adherent to male adult *Megachile rotundata* (Alfalfa leafcutting bees; ALBs). Means and standard error shown.

**Figure 9 insects-11-00359-f009:**
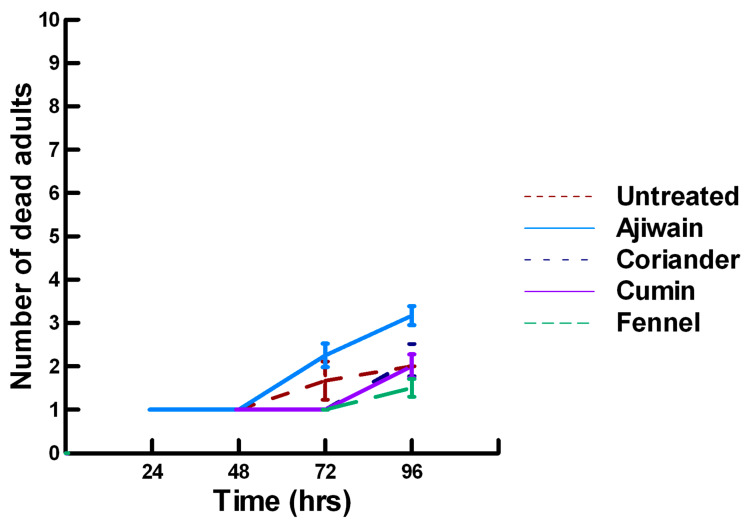
Acute contact toxicity with a low concentration of plant powders from the family Apiaceae (Ajwain, coriander, cumin and fennel) adherent to male adult *Megachile rotundata* (ALBs). Means and standard error shown.

**Figure 10 insects-11-00359-f010:**
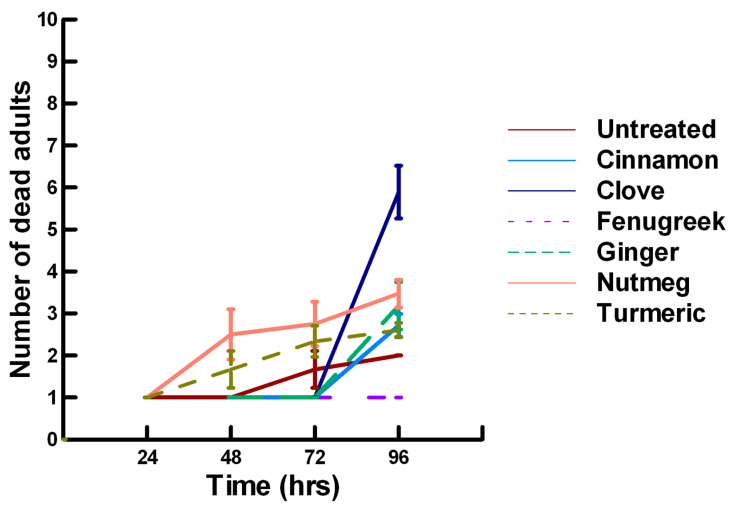
Acute contact toxicity with a low concentration of plant powders from the families Lauraceae (cinnamon), Myrtaceae (clove), Fabaceae (Fenugreek), Myristicaceae (nutmeg) and Zingiberaceae (ginger, turmeric) adherent to male adult *Megachile rotundata* (ALBs). Means and standard error shown.

**Figure 11 insects-11-00359-f011:**
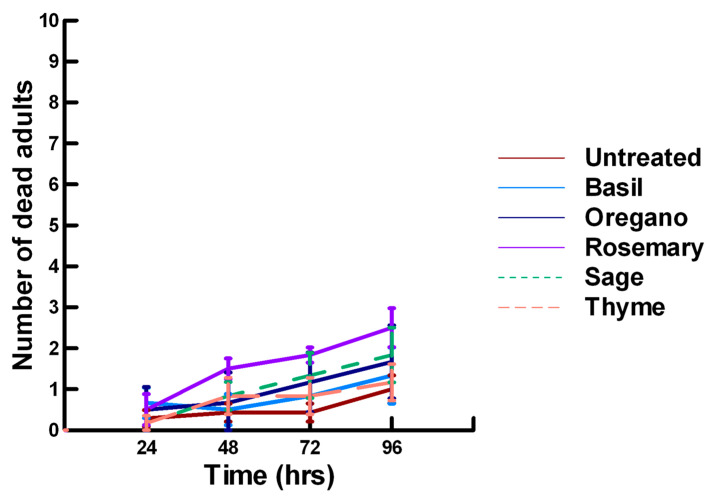
Acute contact toxicity with a high concentration of plant powders from the family Lamiaceae (basil, oregano, rosemary, sage and thyme) adherent to male adult *Megachile rotundata* (ALBs). Means and standard error shown.

**Figure 12 insects-11-00359-f012:**
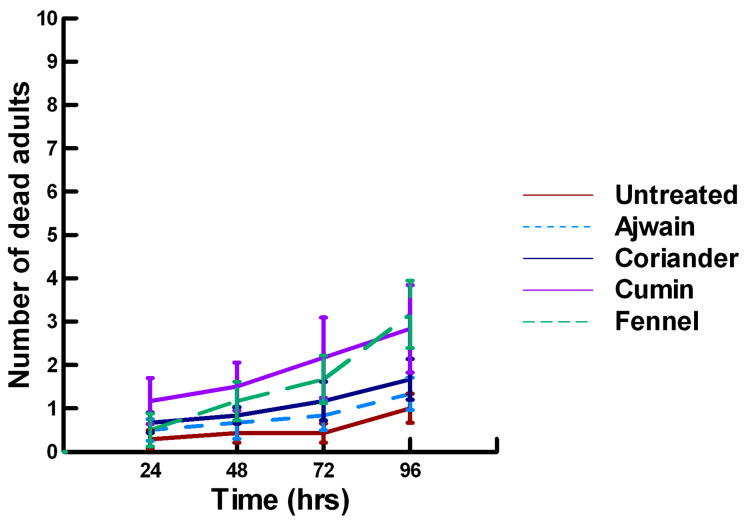
Acute contact toxicity with a high concentration of plant powders from the family Apiaceae (Ajwain, coriander, cumin and fennel) adherent to male adult *Megachile rotundata* (ALBs). Means and standard error shown.

**Figure 13 insects-11-00359-f013:**
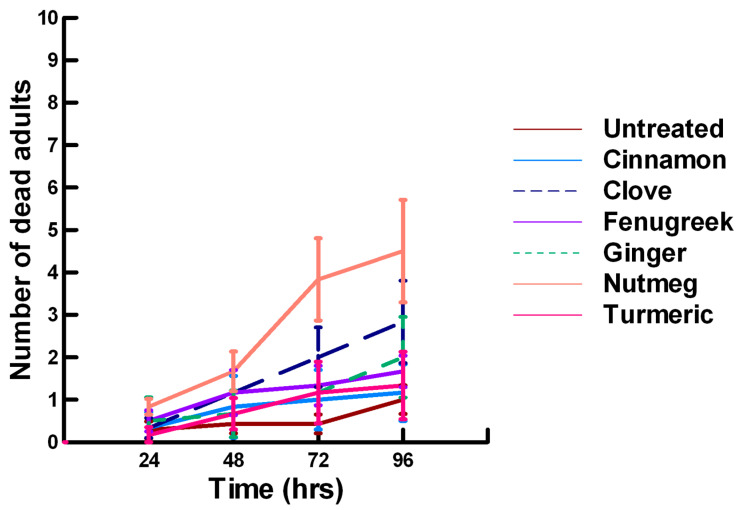
Acute contact toxicity with a high concentration of plant powders from the families Lauraceae (cinnamon), Myrtaceae (clove), Fabaceae (Fenugreek), Myristicaceae (nutmeg) and Zingiberaceae (ginger, turmeric) adherent to male adult *Megachile rotundata* (ALBs). Means and standard error shown.

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
