# Peer review of "Insecticidal Activity of Plant Powders against the Parasitoid, Pteromalus venustus, and Its Host, the Alfalfa Leafcutting Bee"

_insects, 2020, doi:10.3390/insects11060359_

Round 1

Reviewer 1 Report

Review:

Insecticidal activity of plant powders against the parasitoid, Pteromalus venustus, and its host, the  Alfalfa Leafcutting bee

COMMENTS TO THE AUTHOR:

Summary

This manuscript describes the testing of plant powders as an environmentally friendly alternative to chemical insecticides using the parasitoid pest Pteromalus venustus and its host, the agricultural economic important Alfalfa Leafcutting bee (ALB) Megachile rotundata Fab. as model system. Interestingly, the authors found among fifteen tested plant powders nine that caused similar or higher parasitoid mortality, eight of them not harming the ALB. As positive side effect, those effective plant powders are also significantly reducing the offspring of the parasitoids. The results could help finding alternative strategies to the conservative treatment with the organophosphate insecticide Vapona. I find the results of the manuscript stimulating and in a well-written format and I would recommend accepting the manuscript after minor revisions.

General comments:

I realized that in most toxicity tests the graph starts at 24 hrs with 1 as average number of dead adults. Does this reflect the actual numbers or is it a graphical distortion? If it is the latter, I would prefer if the lines would start from zero in terms of time and number of dead individuals. I would also like it better if the number of dead adults could be transformed in percentage of death events. It would have been also interesting and would have made the study more complete if the effect on ALB fertility had been tested, too. For the negative control in the tested assays, I would have favored an inert material with similar particle size (nylon, ceramics, latex beads, bio-inert nano particles,…) and also to include the Vapona insectide as direct one-to-one positive control aside. The alternative negative control would evaluate the effect of non-effective particles on the survival, whereas the Vapona positive control would show a direct comparison to the tested powders. This is just a comment to the authors for future work and is not required for this revision. In the discussion section, a short paragraph about possible risks for human health would be appreciated. For the reproductive effect, is the mating inhibited, the fertilization of the female parasitoid inhibited or are the larvae dying? In the discussion section, a price calculation for the botanicals/plant powder compared to the Vapona insecticide would be desirable.

Specific comments:

-L33 A short sentence that ALBs are lining their nest with cut leaves, therefore their name leafcutting bee, would be nice for an audience not familiar with ALBs.

-L131 Could you please add some more information about the variation in the amounts? How did you assess the particle size? How does the particle size affects the distribution? Is a higher amount correlated with bigger particle size? Or the other way around?  Please specify.

L137 P. venestus in italics.

L1150-152 Please shift the sentence “The exposure concentration…” to L144 after …0.1 g of a plant powder.” for clarity. I am wondering why you make the effort for calculating the right amount for low contact assay and for the high contact assay you just use the 0.1 g plant powder. Could you explain your reasoning for the two different settings?

L256 Please specify the calculated concentrations based on the plant powders used.

L313 Just to be sure, the parasitoids all survived for this reproductive toxicity assay, right?

L434-436 Please expand this sentence with additional information about the correlation particle size (small/big?) on time required for buzz behavior (long/short?). Did you quantify your observations? If yes, this could add some valuable information in the result section.

L503 I suggest to tone down this sentence. You did not test EO of ginger directly to make this statement and you cannot exclude other mode of action.

-price calculation for botanicals

Figure legends: For better visibility, please make the lines thicker for the Figure legends.

Author Response

Review #1:

Insecticidal activity of plant powders against the parasitoid, Pteromalus venustus, and its host, the  Alfalfa Leafcutting bee

COMMENTS TO THE AUTHOR:

Summary

This manuscript describes the testing of plant powders as an environmentally friendly alternative to chemical insecticides using the parasitoid pest Pteromalus venustus and its host, the agricultural economic important Alfalfa Leafcutting bee (ALB) Megachile rotundata Fab. as model system. Interestingly, the authors found among fifteen tested plant powders nine that caused similar or higher parasitoid mortality, eight of them not harming the ALB. As positive side effect, those effective plant powders are also significantly reducing the offspring of the parasitoids. The results could help finding alternative strategies to the conservative treatment with the organophosphate insecticide Vapona. I find the results of the manuscript stimulating and in a well-written format and I would recommend accepting the manuscript after minor revisions.

General comments:

I realized that in most toxicity tests the graph starts at 24 hrs with 1 as average number of dead adults. Does this reflect the actual numbers or is it a graphical distortion? If it is the latter, I would prefer if the lines would start from zero in terms of time and number of dead individuals.

It is not a graphical distortion but is showing the actual number dead. I edited all the figures, to make them easier to interpret, but after setting the time as a continuous variable,  I decided to leave it as a category variable for ease of seeing the different plant powders.

I would also like it better if the number of dead adults could be transformed in percentage of death events. It would have been also interesting and would have made the study more complete if the effect on ALB fertility had been tested, too.

I left the number, because the percentages can be misleading to the reader who is unfamiliar with that concept. In a future manuscript, we evaluate this in a pilot study at commercial locations.

For the negative control in the tested assays, I would have favored an inert material with similar particle size (nylon, ceramics, latex beads, bio-inert nano particles,…) and also to include the Vapona insectide as direct one-to-one positive control aside.

I added a statement to the methods section, see 140-143.  The untreated Pteromalus do have plant powders electrostatically adherent to their bodies from walking through the cocoon matrix. It consists of alfalfa leaf powder. I called it the negative control.

The alternative negative control would evaluate the effect of non-effective particles on the survival, whereas the Vapona positive control would show a direct comparison to the tested powders. This is just a comment to the authors for future work and is not required for this revision.

See above for the negative control. We did not run a Vapona-control because of risk to staff and the results are already published for this chemical and ALBs.

In the discussion section, a short paragraph about possible risks for human health would be appreciated.

I expanded the statement. See line 457-460.

For the reproductive effect, is the mating inhibited, the fertilization of the female parasitoid inhibited or are the larvae dying?

I provided an explanation in the results section, see 316-344.

In the discussion section, a price calculation for the botanicals/plant powder compared to the Vapona insecticide would be desirable.

I did not add this information to the first manuscript as we have not established the dose to apply to the cocoon matrix.   

Specific comments:

-L33 A short sentence that ALBs are lining their nest with cut leaves, therefore their name leafcutting bee, would be nice for an audience not familiar with ALBs.

I added a section. See line 34-36.

-L131 Could you please add some more information about the variation in the amounts? How did you assess the particle size? How does the particle size affect the distribution? Is a higher amount correlated with bigger particle size? Or the other way around?  Please specify.

I expanded the explanation within the methods section. See line 115-123.

L137 P. venestus in italics.

Edited as suggested.

L1150-152 Please shift the sentence “The exposure concentration…” to L144 after …0.1 g of a plant powder.” for clarity. I am wondering why you make the effort for calculating the right amount for low contact assay and for the high contact assay you just use the 0.1 g plant powder. Could you explain your reasoning for the two different settings?

Completed as requested for the moving of the statement. I have clarified the rational within line 115-123.

L256 Please specify the calculated concentrations based on the plant powders used.

Edited as requested, line 253-256.

L313 Just to be sure, the parasitoids all survived for this reproductive toxicity assay, right?

Correct. See line 158-161.

L434-436 Please expand this sentence with additional information about the correlation particle size (small/big?) on time required for buzz behavior (long/short?). Did you quantify your observations? If yes, this could add some valuable information in the result section.

Expanded as requested, line 432-445.

L503 I suggest to tone down this sentence. You did not test EO of ginger directly to make this statement and you cannot exclude other mode of action.

Edited as suggested, See line 514-521.

-price calculation for botanicals

We include this calculation in a future manuscript once concentrations are established for the cocoon matrix.

Figure legends: For better visibility, please make the lines thicker for the Figure legends.

Within the SYSTAT program, I am unable to alter the legend line thickness. I re-set all the figures using different dash and color combinations to ensure that the legend lines/labels could be aligned with the figure.

Reviewer 2 Report

This is a very interesting and useful investigation of the use of plant-derived powders to control a pest of an important managed pollinator.  I have made many remarks on the attached PDF.  Most are aimed at providing clarity for the reader who is unfamiliar with leafcutting bee management. Some clarity is also needed in experiment set-ups.  Also, the methods can be reduced by not repeating methodology, as possible.

The discussion should also include/acknowledge the need to further investigate whether their is an effect of the powders on female bee reproduction or on mating (physical harm or olfactory disruption).  Also, efficacy on their other chalcid wasps might be mentioned. It would also be good if there was a statement and reference for the possible effects of any benign powder that lacked plant volatiles (like unscented talcum powder). 

Author Response

Throughout the text, we changed Alfalfa Leafcutting bee to Alfalfa leafcutting bee. I was not able to confirm that Alfalfa should not be capitalized. 

I did any minor editing requested by the reviewer, as suggested.

Line 31. I did not add the scientific names for the crops.  I left the common names as the list for the scientific names would be long and serve no purpose.

Line 33. I rewrote this section to describe it in a little more detail. 

Line 93. I added the number of cocoons and the box dimensions. 

Line 101. I added the information.

Line 119. I added w/w.

Line 93. I defined the cocoon matrix.

Line 130. Edited as requested.

How did you prevent the filter paper from being soaked by the honey water from the cotton pad?

By adding a limited volume of honey water. It could only penetrate the surface.

Lid? I added that the lid was removed. See 130-133.

How were the six regions represented in each of the 6 replicates?  Or were each of the six regions replicated six times?  Why is the region important for ALBs and/or wasps?

I took out the second number throughout the section. Regional differences in operational practices can affect cocoon quality and by inference ALB and parasitoids.

Line 146. I decided to leave this as it is. Mainly its to enable easier uptake of the method by others without searching for the parts. 

Line 162. Edited for clarification. Generally we tape things down to make sure there is no mortality from physical injury. See 171.

Line 168. This can be described earlier in ms., and simply left as ALB prepupa here. Was the prepua naked, or still in the cell? Edited for clarity. See 181-183.

Line 196. Same as above?  Can just state that.              It is different, so I left it as it is.

Line 212. I defined the variables in the model.

Line 255. What do you mean by tray?  Something from your experiment, or something in bee management?

I think this is easier to understand now with the changes made in the introduction.

Line 258. I have added “ Means and standard error shown”  to the graph titles.

Line 272. Edited as suggested. See 283-284.

Line 316. See line 327. I edited to make it clearer.

Line 324. removed. See line 334.

Line 450. Added the active ingredient. See  line 467.

I do not recall that regions were assessed statistically.  It is not clear in methods how you would have determined a regional effect. And, why would bees from certain regions react differently?

I was not able to analyze this effect because of the experimental design. But the variation or SE, (0-9) dead for the different regions, suggests that the responses vary by region. I did explain this more thoroughly in the results section.

Wasp names: Trichogramma, Braconid etc. I am not sure that these need to be italicized, but I did it based upon the reviewer comment.

If reproduction of the hymenopteran parasitoid is affected by plant powders, might also the female bee's reproduction be affected if she is contaminated upon her emergence in the tray?  This needs further investigation.

Definitely. I added a statement to that effect in the conclusions.

Reviewer 3 Report

General: This manuscript presents valuable information about the control of a parasitoid of alfalfa leafcutter bees.  The authors did an excellent job evaluating the efficacy of numerous plant powders on the parasitoid as well as the impact of the various plant powders on alfalfa leafcutter bees.

Author Response

Specific Comments:

Lines 46-47: “Each female parasitoid can oviposit up to 26 eggs into one cocoon and can lay up to 110 eggs [1] causing the loss of up to 4 developing ALBs.”

Each cocoon contains one ALB. A female parasitoid may lay up to 26 eggs per cocoon and lays up to 110 eggs.  It makes sense that 4 ALBs will be lost per female parasitoid, assuming maximum rates, if each female parasitoid lays the maximum or near the maximum number of eggs per cocoon.  Wouldn’t more ALBs be lost if fewer than 26 eggs were laid by the female parasitoid per cocoon? Does the female parasitoid always lay close to 26 eggs per cocoon?  Do female parasitoids never lay more than 110 eggs? [I was unable to locate the reference so it was not possible to answer these questions.]

“Each female parasitoid can oviposit up to 26 eggs into one cocoon and can lay up to 110 eggs [1] causing the loss of up to 4 developing ALBs.”

The publication if googled can be obtained from

https://publicentrale-ext.agr.gc.ca/pub_view-pub_affichage-eng.cfm?=undefined&publication_id=1495E&wbdisable=true

I replaced it with another reference to the same information that is available through another article.

Lines 37-38, 48, 52: 

What does this mean? 

“The male bees emerge between day 18 and 28, while the female bees emerge between day 21 and 28 [3].”

  • When do days 18-28 occur? When do days 21 and 28 occur? When does “day 8 or 9” occur?  
  • When is day zero? What are these days counting? Is there a day length, temperature, etc. that triggers day 1? Please explain the chronology. Do not presume that the reader knows ‘degree days’, etc. When is diapause broken?  What is the temperature of diapause? What is the temperature on Day 0 and Day 1?
  • What is going on with ALBs at this time? What is going on with the parasitoid?

I edited this section to indicate in general terms, the sequence for incubation. See line 39-40. I did not elaborate on how the process affects diapause for either insect as this is not a physiology manuscript. I did edit the parasitoid section to clarify the number of eggs laid and the outcome. See line 48-49.

Lines 47-51: It may not be necessary to describe this process of the second generation of the parasitoid in such detail. What is important is the second generation of the parasitoid emerges before ALBs emerge and this difference in emergence results in an amplification of ALB loss from parasitism.

I left this information as it explains why we measured the number of first and second generation parasitoids.  

Lines 56-71: This paragraph is confusing. From this paragraph it is not clear if plant powders and essential oils from plant powders are different. Do the essential oils made from specific parts of the plant differ from essential oils made from plant powders? 

Lines 56-59: An example of the confusion results in the following suggested rewording:

“Many studies have evaluated the insecticidal properties of… plant “essential oils from seeds, leaves, flower buds, flower heads, rhizomes, bark,…” peels, or powders “…to protect stored products from a variety of pest insects [1129].”

Edited as suggested. I removed the essential oil component. See line 58.

Lines 83-86: It is argued that males ALB were tested rather than female ALB because males are the first to emerge and have a greater exposure period.  This argument would be strengthened if the potential insecticides tested had a short period of efficacy.  Is there any information about the half-life of the tested materials?

We discussed the difference in stability of the released EOs within the discussion as they relate to plant powders vs EOs (See line 463-464).  

Line 93:

Is day 15 part of the 30 days in which the cocoons “maintained at 28 °C, 16:8 92 photoperiod and 50% relative humidity”? Please clarify.

Edited as suggested.

Line 95: It is correct to assume that days 17-24 are also part of the same 30 days?  Please clarify.

Edited as suggested.

Line 95: When was Day 1 of the experiment?  Was it before or after the 30 days? Please clarity.

Edited as suggested.

Lines 95-97: Were the males removed immediately prior to the bioassay? If so, please make this clear.  As written, this sentence could be understood to say that the male ALBs were held for some unknown length of time before the assays were conducted.

“On the day of the experiment, ten adult male bees were transferred using feather light forceps into each 50 ml Fisherbrand™ conical tube until use in the bioassays.”

Edited as suggested.

Line 101-102: It is correct to assume that it was an unknown number of males and females that were added?  If so, the following wording is suggested:

“From 30 to 50 parasitoids, male to female ratio unknown, were added to each container, …”

Edited as suggested.

Lines 99-105: Does this section describe how a sufficient number of parasitoids were obtained for the bioassay?  If so, please say so. The section header could say “Rearing of Parasitoids” or the text could be altered to make it clear that these parasitoids were used for the bioassay (as was done in the previous section).

Edited as suggested.  I also changed the headings as suggested for 2.1 and 2.2.

Lines 103-105: Why were the parasitoids collected?

“Adult P. venustus were collected using an aspirator system (Tetra® Whisper 60 Air Pump with tubing connectors attached to a Fisherbrand™ specimen container) between day 12 and 15.”

Edited as suggested.

Lines 114-121: No data or citations are given for the claim that “…adult parasitoids could come into contact with either a sparse particle concentration from emergence and walking within the cocoon matrix, or a heavy particle concentration from settled plant powders within the cocoon matrix.” 

Provide evidence for this claim. This is critical as these two exposure categories are critical for the design of the study.

See line 116-124. We performed a preliminary study to determine how plant powders behaved within the cocoon matrix. Two types of exposure were observed as indicated within the paragraph. This was not known for either the bee or the parasitoid.

Lines 138-140: Assuming that no difference in parasitoid weight was observed, this sentence should be changed from: “No weight changes were recorded…” to No weight changes were observed…”

Edited as suggested.

Line 153: Assuming that no difference in parasitoid weight was observed, this sentence should be changed from: “No weight changes were recorded…” to No weight changes were observed…”

Edited as suggested.

Edited as suggested.

Lines 189, 204, 219, 222, 229, 230, 233, 236, 238, 240, 241, 245, 246, 248, 273-275, 277, 280, 282, 285, 290, 292, 309, 316, 319, 322, 328, 329, 332, 335, 338, 354, 355, 357, 358, 361, 356, 368, 373, 377, 394, 398, 399, 401, 403, 405, 407, 412, and 415 and Figures 1s–6 and 8–13 :

Please use the word ‘mean’ rather than ‘average’, assuming this is the descriptive statistic being used.  The definition of average includes mean, median or mode.   

Edited as suggested.

Figures 1–6 and 8–13: It is hard to match the color of the lines in the key to the lines in the graph. Two suggestions: It would be useful if 1) the lines in the key were broader and 2) the order of the key matched the order of the lines in the graph, as much as possible. [Maybe the order at the 96 hours as the spread of the lines are (mostly) greatest at this time.]

This was a challenge. I tried several approaches to improve the visibility. In the end, I decided to go with a combination of color and symbols. I hope that this addresses your concerns.

Lines 249-251: What criteria are you using to rank the materials tests? Are you using the mean number of dead parasitoids, the z-score, the p-value for the z-score, some combination of these three criteria, “achieving >60% mortality at 24 hrs.” or some other criterion?  Depending upon the criterion, the hierarchy will differ.

None of the methods I can deduce from the manuscript match the ranking. [“The hierarchy for the fumigant properties associated with the plant powders under low contact conditions is Ajwain, cinnamon, clove, fennel, nutmeg, turmeric > cumin, ginger > oregano.”]  

It is a common practice for receptor physiology as it relates to the mode of action of these materials. It is a relative comparison. I have edited this to indicate that it is relative activity.

Are you considering Ajwain, cinnamon, clove, fennel, nutmeg and turmeric equal in efficacy?

Are you considering cumin and ginger equal in efficacy? Why?

Please describe how you are ranking the materials tests. Provide a justification for the ranking method chosen.

The hierarchy of the materials tested under low contact conditions, using the mean number of dead parasitoids is:

Clove (10, 13.474 P<0.0002) = fennel (10, 13.390, P<0.0002) > Ajwain (9.7, 12.455, p<0.0002) > nutmeg (9.6, 11.939, P<0.0002) > cinnamon (9.1, 10.805, P<0.0002) = turmeric (9.1, 10.185, P<0.0002) > ginger (7.9, 5.851, P<0.0002) > cumin (7.1, 4.598, P<0.0002) > oregano (6.1, 2.023, P=0.0217),

The hierarchy of the materials tested under low contact conditions, using the z-score is: Clove (10, 13.474 P<0.0002) = fennel (10, 13.390, P<0.0002) > Ajwain (9.7, 12.455, p<0.0002) > nutmeg (9.6, 11.939, P<0.0002) > cinnamon (9.1, 10.805, P<0.0002) > turmeric (9.1, 10.185, P<0.0002) > ginger (7.9, 5.851, P<0.0002) > cumin (7.1, 4.598, P<0.0002) > oregano (6.1, 2.023, P=0.0217)

The hierarchy of the materials tested under low contact conditions, using the p-value is: Clove (10, 13.474 P<0.0002) = fennel (10, 13.390, P<0.0002) = Ajwain (9.7, 12.455, p<0.0002) = nutmeg (9.6, 11.939, P<0.0002) = cinnamon (9.1, 10.805, P<0.0002) = turmeric (9.1, 10.185, P<0.0002) = ginger (7.9, 5.851, P<0.0002) = cumin (7.1, 4.598, P<0.0002) > oregano (6.1, 2.023, P=0.0217)

Lines 254-256: Suggested rewording:

To mimic the concentration used in the low contact studies, the amount of plant powders required per tray, using the cocoon weight occupying the tray volume, was calculated to be equal to 1 to 6% concentration, depending upon the plant powder.

Edited as requested.

Lines 272-273: It might be useful to use the phrase “dose-response relationship”. Suggested rewording:

Increasing the contact concentration of the plant powders did not alter the dose response relationship observed for intoxication as the length of exposure increased.

Edited as requested.

Lines 293-294: Please describe how you are ranking the materials tests. Provide a justification for the ranking method chosen. See comments above for Lines 249-251.

This has been addressed throughout the text.

Lines 309-312: This information belongs in the Discussion section.

I removed the information not relevant to the results section.

Line 316: Please explain how “the treatment average” was determined. 

From the text, in the untreated parasitoid group, a mean of 2 immature bees were parasitized per female parasitoid. It is then stated that among the various treated parasitoid groups, the mean number of immature bees parasitized by parasitoids was 0.36.  How was this number determined?  Why was this value used rather than comparing the treatments to the untreated group?

The treatment mean represents the effect on a measured count variable and this is compared with the effects of individual treatments. I added this as part of the description in the statistics section.

Figure 7: Please indicate which treatments differed significantly from the untreated controls.

Edited as requested.

Line 322-324: This sentence belongs in the Discussion section.

Edited as suggested.

Lines 328-330: Why are the ‘number of immature bee attacks’ being discussed? This section is describing “the number of parasitoid larvae developing from each parasitized immature bee.”  Should not the sentence read as follows:

“Ginger applications (1.62, -.508, P=0.30503) elicited an equivalent number of parasitoid larvae development from each parasitized immature bee as the treatment average or 91% lower than with the untreated parasitoids.”

I edited this section to indicate that it is the number of parasitoid larvae.

Lines 330-333 and 333-336: See comment above.

Edited as suggested.

Lines 338-341: The following wording is suggested:

In contrast, Ajwain, cumin, fennel and ginger weakly reduced (~45%) the fertility of the female parasitoids (14.0, 12.7, 17.3, and 13.9 parasitoids per bee larvae, respectively), while cinnamon, clove, oregano and turmeric  strongly reduced (~85%) the fertility of the female parasitoids (3.4, 0.0, 0.0 and 6.2 parasitoids per bee larvae, respectively).

I replaced the statement with the suggested sentence.

Lines 342-344: This sentence belongs in the Discussion section.

Edited as suggested.

Line 359-360: Considering moving the following phrase to the Discussion section:

“…suggesting that there may be a genetic or management component to the regional differences in bee sensitivity.”

I did not remove this statement as it provides a possible reason for the observed differences that we were not able to explore in this study.

Lines 368-369: Consider moving the following phrase to the Discussion section:

“…warranting further investigation prior to eliminating this plant powder as a candidate for controlling the parasitoids.”

I did not remove this statement as it provides an indication that even the best parasitoid control agents may pose a significant risk for the bees.

Lines 378-379: Please describe how you are ranking the negative effects of plant powders on ALB. Provide a justification for the ranking method chosen.  See comments for Lines 249-251.

“The hierarchy for the negative effects of plant powders on adult ALBs is nutmeg, rosemary> clove.”

Relative potency, as indicated above.

Lines 379-380: This sentence belongs in the Discussion section.

Edited as suggested.

Lines 396-397: This sentence belongs in the Discussion section.

“This same trend was observed for the low contact with plant powders.”

Removed from the text.

Lines 408-409: This sentence belongs in the Discussion section.

“These plant powders should not be applied at concentrations where adult bees can encounter accumulated material

within the cocoon matrix.”

Removed from the text.

Lines 413-416: The following wording is suggested:

“In contrast, one member of the family Myristicaceae (nutmeg, 4.5, 6.711, P<0.0002) and One member of the family Myrtaceae (clove, 2.8, 2.028, P=0.0212) significantly increased adult bee mortality compared with the treatment average and these losses are above those reported with Vapona® applications [8].”

Edited as suggested.

Lines 416-444: These sentences belong in the Discussion section.

I left the information in the results section as it is relevant to the outcomes.

Line 462-463: No information in the results section could be found to support the following:

“…while clove poses a high risk for adult ALBs from specific regions”

Either delete this phrase or provide data in the Results section.

I expanded the explanation for this in the results section. See line 366-367.

Lines 465-476: The justification for using plant powders rather than essential oils is clearly explained.  What is the potential for the quantity of the toxic compound in the plant powder to vary? As the dose to which both the parasitoid and ALB are exposed is critical for pest control efficacy and the lack of non-target toxicity, respectively, if the quantity of the toxic compound varies substantial the pest control efficacy could be lost or the adverse impact on ALB could increase.

Edited as suggested. See line 469-474.

Line 490: Start a new paragraph with the sentence beginning: “In the current study, members…”

Edited as suggested.

 Line 500: Start a new paragraph with the sentence beginning: “There is one study…”

Edited as suggested.

Line 507: Start a new paragraph with the sentence beginning: “There is no information…”

Edited as suggested.

The text, Line 505, alludes to the potential impact on the efficacy and non-target effects of plant powders depending upon the plant species of ginger.  This idea should be explored more thoroughly for any of the plant powders for which difference species exist.  For example, over 250 species in the cinnamon genus exist.  Which of this might go by the common name cinnamon and could be used as a plant powder but not show the same efficacy and low non-target species impact?

Edited as suggested. I added a statement to the discussion. See 469-475. In our own hands, we have implemented these on-farm by pre-testing the powders for efficacy prior to the spring incubation. The farmers have the capacity to do this themselves, and over time the idea is to have a list of suppliers that are “safe” for the bees.

Potential future work: It would be interesting to know the actual mechanisms of action for both toxicity and reproductive impact of the various plant powders tested.

Absolutely. I added this to the conclusion.